# Synthesis, Experimental and Theoretical Study of Azidochromones

**DOI:** 10.3390/molecules27092636

**Published:** 2022-04-20

**Authors:** Ena G. Narváez-Ordoñez, Kevin A. Pabón-Carcelén, Daniel A. Zurita-Saltos, Pablo M. Bonilla-Valladares, Trosky G. Yánez-Darquea, Luis A. Ramos-Guerrero, Sonia E. Ulic, Jorge L. Jios, Gustavo A. Echeverría, Oscar E. Piro, Peter Langer, Christian D. Alcívar-León, Jorge Heredia-Moya

**Affiliations:** 1Facultad de Ciencias Químicas, Universidad Central del Ecuador, Francisco Viteri s/n y Gilberto Gato Sobral, Quito 170521, Ecuador; egnarvaezo@uce.edu.ec (E.G.N.-O.); kevincarcelen@outlook.com (K.A.P.-C.); dazuritas@uce.edu.ec (D.A.Z.-S.); pmbonilla@uce.edu.ec (P.M.B.-V.); tgyanez@uce.edu.ec (T.G.Y.-D.); 2Centro de Investigación de Alimentos CIAL, Universidad UTE, Quito 170527, Ecuador; luis.ramos@ute.edu.ec; 3CEQUINOR (CONICET-UNLP), Facultad de Ciencias Exactas, Universidad Nacional de La Plata, Bv. 120 No 1465, La Plata 1900, Buenos Aires, Argentina; sonia@quimica.unlp.edu.ar; 4Departamento de Ciencias Básicas, Facultad de Ciencias Exactas, Universidad Nacional de Luján, Rutas 5 y 7, Luján 6700, Buenos Aires, Argentina; 5Laboratorio UPL (UNLP-CIC), Camino Centenario e/505 y 508 (1897) M.B. Gonnet and Departamento de Química, Facultad de Ciencias Exactas, Universidad Nacional de La Plata, República Argentina. 47 esq. 115, La Plata 1900, Buenos Aires, Argentina; jljios@quimica.unlp.edu.ar; 6Departamento de Física, Facultad de Ciencias Exactas, Universidad Nacional de La Plata e IFLP (CONICET, CCT-La Plata), La Plata 1900, Buenos Aires, Argentina; geche@fisica.unlp.edu.ar (G.A.E.); piro@fisica.unlp.edu.ar (O.E.P.); 7Institut für Chemie, Universität Rostock, Albert-Einstein-Str. 3a, 18059 Rostock, Germany; peter.langer@uni-rostock.de; 8Leibniz Institut für Katalyse, Universität Rostock e. V. (LIKAT), Albert-Einstein-Str. 29a, 18059 Rostock, Germany; 9Centro de Investigación Biomédica (CENBIO), Facultad de Ciencias de la Salud Eugenio Espejo, Universidad UTE, Quito 170527, Ecuador

**Keywords:** azidochromone, spectroscopic properties, structural X-ray diffraction, quantum chemical calculations, Hirshfeld surface analysis

## Abstract

A series of 2-(haloalkyl)-3-azidomethyl and 6-azido chromones has been synthetized, characterized and studied by theoretical (DFT calculations) and spectroscopic methods (UV-Vis, NMR). The crystal structure of 3-azidomethyl-2-difluoromethyl chromone, determined by X-ray diffraction methods, shows a planar framework due to extended π-bond delocalization. Its molecular packing is stabilized by F···H, N···H and O···H hydrogen bonds, π···π stacking and C–O···π intermolecular interactions. Moreover, AIM, NCI and Hirshfeld analysis evidenced that azido moiety has a significant role in the stabilization of crystal packing through weak intermolecular interactions, where analysis of electronic density suggested closed-shell (CS) interatomic interactions.

## 1. Introduction

Chromone derivatives with reactive functional groups such as halogens [1,2], haloalkyl [3,4,5], hydroxyl [6], amino [7,8] and azide [8,9] are important cores and precursors for pharmacologically active compounds (anti-cancer, anti-HIV, antioxidant, anti-tubercular, anti-inflammatory, analgesic, antimicrobial, anti-malaria, gastroprotective, antihistamine and antihypertensive) [10,11,12,13,14]. Among them, haloalkyl groups and azides are particularly relevant, both for their accessible synthetic routes and for their wide use as precursors of reactive species [15]. Nitrenes and nitrenium ions can be generated from azides, which also serve as anchors for synthesis of nitrogen-rich compounds such as aziridines, azirines, triazoles, triazolines and triazides [9,15].

Different 2-trifluoromethyl and 2-(polyfluoroalkyl)chromones have been studied extensively [16,17,18,19]. Despite their ready accessibility, these compounds have been ignored by synthetic chemists, although their systematic study has been revitalized in recent years. The main basic structure of chromones serves as a scaffold for novel compounds with biological importance, improving their solubility and pharmacokinetic properties [20,21,22]. In addition, the presence of electron-withdrawing polyfluoroalkyl groups attached to the C-2 carbon of chromones crucially modifies the reactivity and electron-density distribution around the α,β-unsaturated carbonyl portion of the pyrone ring [23]. These changes expand the synthetic potential of 3-substituted-2-polyfluoroalkylchromones, their structural study being our area of interest in this research.

This work reports the synthesis, characterization and computational study of five novel halo-chromones (Figure 1, Appendix A and Appendix A) with the azido group.

## 2. Results and Discussion

### 2.1. Conformational Analysis

Geometries of compounds **1**–**5** in the gas phase were studied, determined and calculated through a conformational analysis using the B3LYP/6-311++G(d,p) level of theory. Potential energy curves were performed by torsions around the C2–C2′ and C3–C3′ bonds that connect both exocyclic carbons C2′ (bearing the halogen atoms) and C3′ with the corresponding C2 and C3 carbon atoms of the heterocyclic ring C2 and C3, respectively. The most stable conformations of **1**–**5** and the main parameters are presented in Table 1 (See Appendix A).

### 2.2. NMR Spectroscopy

Experimental chemical shifts were compared with those derived from chemical calculations using the B3LYP/6-311+G(2d,p) level of theory (Table 2). For protons, a good agreement is observed with Δ = TM_exp_ − TM_calc_ deviation ranging from −0.49 to 0.27 ppm, except for the CH_2_N_3_ of **1** (Δ = −0.74). For carbon, the deviation range is between −32.2 and 10.9 ppm. The linear relationship between computed and experimental data, for all compounds, gives R-square values above 0.978 and 0.945 for protons (see Appendix A) and carbons, respectively. The highest deviation was found for CF_2_Cl and CF_3_ of **1** and **2** with Δ values of −32.2 and −22.1 ppm, respectively; this suggests that the isotropic shielding of the fluorine atoms is underestimated by theoretical calculations, as previously reported [24,25,26,27]. Analysis of signal multiplicity and coupling constants was used, in conjunction with the calculated data, for structural elucidation.

### 2.3. Electronic Spectra

The calculated and experimental (using methanol as solvent (**1**: 1.05 × 10^−5^ M, **2**: 1.25 × 10^−4^ M, **3**: 1.05 × 10^−5^ M, **4**: 1.99 × 10^−5^ M, **5**: 2.51 × 10^−5^ M)) electronic absorption spectra (see Appendix A) of **1**–**5** are shown in Table 3. Electronic spectra were calculated at the TD-B3LYP/6-311++G(d,p) level of theory, implicitly considering the influence of the solvent (methanol, ε = 32.7), with the conductor-like polarizable continuum model (CPCM). The main experimental absorption bands were correlated with the calculated vertical electronic transitions, with oscillator strengths (f) > 0.076.

The main molecular orbitals, involved in the electronic transitions of **1**–**5**, are depicted in Appendix A.

The most intense absorption band of **1**, localized at 204 nm (calc. 211 nm), is attributed to HOMO − 1 → LUMO + 3 transitions principally due to excitations from π-bonding orbitals of the aromatic ring and non-bonding orbitals of N1 and N3 atoms to π*-orbitals of the benzene ring and p-type orbitals of one fluorine atom.

The strong bands at 203 nm for **2** and 204 nm for **3***–***5** derive from electronic excitations from HOMO → LUMO + 5 (**2**) and HOMO − 2 → LUMO + 2 (**3***–***5**). The absorption at 203 nm (calc. 212 nm) arises from principally π-bonding orbitals of the aromatic ring and nitrogen atoms (N1 and N3) non-bonding orbitals of the azide moiety to π*-orbitals delocalized along the whole molecule.

The absorption at 204 nm (**3***–***5**) is originated in all cases from π → π* transitions within the aromatic ring and to π*-orbitals of C=C and N2=N3 bonds.

The dominant HOMO → LUMO excitation for **1**–**5** was assigned to the observed bands at 303, 318, 305, 301 and 304 nm, respectively. This absorption is mainly generated by transitions from π-orbitals of the aromatic ring and C=C bonds and with the contribution of non-bonding orbitals of both oxygen, N1 and N3 atoms to π*-orbitals of the benzene ring and p-type orbitals of some carbon atoms of the heterocycle.

### 2.4. Crystallographic Structural Results of (**4**) or: Crystal Structure of **4**

Figure 1 is an ORTEP [28] drawing of the molecule, and its bond distances and angles are presented in Appendix A.

Due to extended π-bonding, the chromone molecular skeleton is planar (rms deviation of atoms from the best least-squares plane is 0.043 Å). Observed bond distances and angles agree with established organic chemistry rules. In fact, phenyl ring C–C distances [from 1.363(3) to 1.398(3) Å] are as expected for a resonant-bond structure. The fused heterocycle shows a C2–C3 bond length of 1.339(3) Å, shorter than the other heterocycle C–C distances [from 1.381(3) to 1.468(3) Å], which corresponds formally to double-bond character for that link. Heterocycle C–O single-bond distances are 1.355(2) Å and 1.375(2) Å, and the carbonyl C=O double-bond length is equal to 1.227(2) Å. The molecular metrics of the chromone skeleton agree with one of the corresponding closely related chromone derivatives [5,24,25]. The –C(_sp3_)F_2_H and –C(_sp3_)H_2_(N_3_) groups exhibit the expected tetrahedral bonding structure. C–C–F angles are 108.1(2)° and 110.3(2)° and the F–C–F angle is 106.0(2)°. The C–C–N_3_ bond angle is 111.23° and ∠ (C–N=N2) = 114.5(2)°. The observed C–F bond distances are 1.351(2) and 1.363(3) Å and d(C–N3) = 1.478(3) Å. Azide N=N bond lengths are 1.221(2) Å (bonded to the carbon end) and 1.126(3) Å.

### 2.5. Hirshfeld Surface Analysis

Figure 2a shows the crystal packing of **4** stabilized by $R_{2}^{2}\left(14\right)$, $R_{4}^{4}\left(8\right)$,$\;R_{2}^{2}\left(10\right)$, $R_{4}^{4}\left(8\right)$ and $R_{2}^{2}\left(13\right)$ graph-set ring motifs, which form dimers and tetramers in the supramolecular assembly. The F1 and F2 fluorine atoms of the –CF_2_H moiety interact individually with an aromatic and an aliphatic hydrogen atom, forming C7_(sp2)_–H7···F2 and C3′_(sp3)_–H3′···F1 intermolecular contacts along the c-axis. Moreover, the C7_(sp2)_–H7···F2 angle is nearly linear (173°), while the observed angle for C3′_(sp3)_–H3′···F1 (107°) reveals a moderate angular directionality. In this sense, the d(H7···F2) intermolecular distance (2.540 Å) is shorter than that observed in 3-bromomethyl-2-trifluoromethylchromone [3] and CF_3_/CF_2_H-substituted benzene, where the role of CF_3_/CF_2_H groups in the C–H···F intermolecular interactions contributes significantly to the stability and molecular arrangement of the crystal structures (see Appendix A). While the C2′_(sp3)_–H···O2, C5_(sp2)_–H5···N1, C2′_(sp3)_–F1···N2 and C2′_(sp3)_–F2···N3 contacts could be considered weak intermolecular interactions due to their directionality and length [29], the azide (N_3_), carbonyl and CF_2_H groups play a significant role in crystal packing. This becomes evident by the relative contribution percentage of the intermolecular contacts, the character and their interaction energies evaluated with the Hirshfeld Surface, AIM and NCI analysis (see below).

Intramolecular π-stacking contacts (Figure 2b) are observed between the chromone ring (Cg1···Cg3, distances of 3.674 Å and Cg2···Cg3, distances of 3.569 Å) and the benzene moiety, and intramolecular offset-face-to-face π-stacking contacts between the benzene moiety and 4-pyrone ring (Cg1···Cg2, distances of 3.594 Å). Moreover, short distances for π-stacking contacts between the chromone rings (Cg3···Cg3, distances of 3.674 Å) and π···π arrangements in **4** show short distances of contact (Appendix A) compared with chromone derivatives substituted in 3, 5 and 7 positions [30], where the linear stacking of forces π···π can be considered the agent that influences the strengthening of the molecular assembly. Additionally, Figure 2c shows C–O···π contacts between the carbonyl group and 4-pyrone ring. The oxygen atom of the carbonyl group in the C-O···π intermolecular interactions (Appendix A) behaves similar to interactions observed in N-arylamides, where the interactions evidence a stronger electrostatic character [31].

To explore the features associated with the role of intermolecular contacts, a Hirshfeld surface analysis of **4** was performed (Figure 3 and Figure 4) [32]. Figure 3a evidences contacts shorter than van der Waals, as highlighted by the red dots on the d_norm_ surface, where the hydrogen and fluorine atoms of the –CH_2_N_3_, –CF_2_H and –C_Ar_–H moieties promote intermolecular contacts. In this sense, Appendix A show the crystal lattice energy calculated with the CE-B3LYP/6− 31 G(d,p) energy model; for the C–H···F contacts, a character significative of contribution-dispersive (E_dis._) and -repulsive interaction (E_rep._) energies was observed. Particularly, the C3′-H3′A···F2 intermolecular contact displayed an important contribution with a relative high impact of dispersive, repulsive and total energy of −23.4 kcal/mol, 12.9 kcal/mol and −21.2 kcal/mol, respectively. The interactions’ energy results are in agreement with those observed in 3-dibromomethyl-2-difluoromethylchromone [5]. Moreover, shape index and curvedness properties evidence π···π stacking and C–O···π interactions [33], which arise from planar stacking arrangements between the chromone rings. Features such as ‘bow-tie’ patterns of large red and blue triangles (see red circle in Figure 3b) and large green flat regions delineated by a blue outline (Figure 3c) reveal these close contacts associated with weak interactions [30].

Figure 4 shows the 2D-fingerprint plot of the main intermolecular contacts of **4**. The pair of narrow spikes, labeled 1, corresponds to the shortest F···H distance associated to C–H···F interactions, while 2 evidences the N···H contacts that arise from C5–H5···N1 interactions. The F···H and N···H contacts have major contributions (23% and 22%) due to the relatively high proportion of fluorine and nitrogen atoms interacting in the crystal structure. Moreover, related chromones with –CF_3_, –CF_2_H moieties revealed a high proportion of weak F···H contacts that provide stability to the crystal structures [3,25]. On the other hand, labels 3 and 5 show H⋅⋅⋅H and C⋅⋅⋅C contacts with lower percentages of relative contribution (14% and 9%, respectively), related to π···π and C–H···π arrangements. Moreover, a moderate percentage of O⋅⋅⋅H (10%) hydrogen bonds, due to the oxygen atom of the carbonyl group and hydrogen of benzene moiety [30], is observed. A similar percentage of relative contribution was evidenced for N⋅⋅⋅F (8%) and C⋅⋅⋅O (6%) contacts with labels 6 and 7, whose characters will be discussed later through AIM and NCI analysis.

### 2.6. AIM and NCI Analysis of Intermolecular Contacts of **4**

Figure 5 shows the self-assembled tetramer of compound **4**, where the theory of atoms in molecules (AIM) has been included by visualization of the noncovalent interactions (NCI) by means of the critical points (CPs) and bond paths [34]. Seven CPs were taken into account in order to understand and reveal the weak interactions, where the reduced density gradient (RDG) [35] is visualized with a color. Isosurfaces of the H-bond and halogen bond (blue color), van der Waals interactions (green color) and strong repulsion areas (red color) are shown. In this sense, the seven CPs (1–7) illustrate vdW interactions, where the azide group and their three nitrogen atoms play a significant role, with large RDG isosurfaces (green color), and participate in several N⋅⋅⋅H and N⋅⋅⋅F contacts [36]. On the other hand, the combined AIM/NCI plot and topological parameters (Table 4) were calculated using the Multiwfn program [37], considering the main (3, −1) CPs according to Bader´s theory of AIM [34]. Therefore, the topological criteria of electron density [$\rho \left(r\right)$] and Laplacian of the electron density $\left[\nabla^{2}\rho \right]$ showed values that suggest a weak interaction [$\rho \left(r\right):\;$0.0035 − 0.0067; $\nabla^{2}\rho$: 0.0151 − 0.0295], according to what is proposed by Koch and Popelier [38] to provide information on the nature of interactions. Likewise, the above-mentioned findings are in agreement with Rozas et al. [39], who state that the strength of the hydrogen bond could be evaluated by the sign of $\left[\nabla^{2}\rho \right]$ and $H\left(r\right)$ as total electronic energy density, where weak interactions show values of $\left[\nabla^{2}\rho \right]$ and $H\left(r\right)$ > 0 (energy interaction < 12.0 kcal/mol). Moreover, according to Espinosa et al. [40], the $\left|\frac{-V\left(r\right)}{G\left(r\right)}\right|$ indicator suggests, for all intermolecular contacts, a weak closed-shell (CS) interatomic interaction, according to the values of Table 4 $\left(\left|\frac{-V\left(r\right)}{G\left(r\right)}\right|<1\;and\;Hr>0\right)$.

## 3. Experimental

### 3.1. General

All solvents and reagents were from Aldrich (St. Louis, MO, USA) and used without further purification. The melting points (uncorrected) were determined on a Büchi Melting Point M-560 (Büchi Labortechnik AG, Flawil, Switzerland). Infrared absorption spectra (KBr disk) were recorded on Varian 660-IR FT-IR (Agilent Technologies, Santa Clara, CA, USA) spectrometer with 2 cm^−1^ of resolution in the range of 4000–400 cm^−1^ (see Appendix A). The Raman spectra of the solid were performed in the range of 3500–100 cm^−1^ at room temperature on a Thermoscientific DXR Raman microscope (Thermo Scientific, Madison, WI, USA) using a diode-pumped solid-state laser of 780 nm, with spectral resolution of 5 cm^−1^ (see Appendix A). The ^1^H-, ^19^F- and ^13^C-NMR spectra were recorded at 25 °C on a Bruker Avance II 500 spectrometer (Bruker, Billerica, MA, USA) using CDCl_3_ and CD_3_OD as solvent. Chemical shifts (δ) are expressed in ppm relative to TMS for ^1^H- and ^13^C-NMR and TFA (δ = −71.0 ppm) for ^19^F-NMR (see Appendix A. Absorbance was measured on a Varian 50 BIO UV-Visible Spectrophotometer (Agilent Technologies, Santa Clara, CA, USA) at 2.0 nm spectral bandwidth using methanol as solvent (see Appendix A. GC-MS spectrometry for 1–3 and 5 were obtained using an Agilent Technologies 7890A chromatograph an 5975C mass-selective detector. The electron energy was 70 eV with a mass range of 50–500 amu and a pressure in the mass spectrometer lower then 10^−5^ Torr. The mass spectra are shown in Appendix A. Mass spectrum of 2 was obtained with a LC-ESI-MS spectrometer UHPLC-MS/MS (model XEVO TQ-S, Waters, Milford, MA, USA) with an electrospray (ESI) interface operating in the positive mode. Reactions were monitored by TLC on silica gel using ethyl acetate/hexane mixtures as a solvent and compounds visualized by UV lamp. The reported yields are for the purified material and are not optimized.

### 3.2. Synthesis: General Procedure for Azidochromones (**1**–**5**)

All bromine-substituted 3-polyhaloalkylchromones were prepared and purified by the reported methods [4,5]. 

The bromine-substituted 3-polyhaloalkylchromone (0.33 mmol), sodium azide (1.26 mmol) and acetone (10 mL) were added to a 50 mL bottom flask with ground glass joint (Figure 2). The reaction conditions for each compound are described in the Appendix A. The end of reaction was monitored by TLC (hexane-EtOAc, 9:1). The mixture was filtered and washed with cold acetone, then crude product was dried under vacuum using a rotary evaporator and recrystallized in hexane to yield the pure compound.

*3-Azidomethyl-2-chloro(difluoro)methylchromone* (**1**). White crystalline solid; yield 97%; mp: 70.5–73.6 °C; UV (MeOH): 204, 225, 244, 303 nm; IR (KBr): 2172, 1649, 1609, 1104, 627 cm^−1^; Raman: 2123, 1650, 1609, 671 cm^−1^; ^1^H-NMR (500 MHz, CDCl_3_) δ: 8. 27 (d, *J* = 8 Hz, 1H, H-5), 7.82 (t, *J* = 8.5 Hz, 1H, H-7), 7.59 (d, *J* = 8.5 Hz, 1H, H-8), 7.53 (t, *J* = 8 Hz, 1H, H-6), 4.53 (t, *^5^J*_H,F_ = 2 Hz, 2H, CH_2_N_3_); ^13^C-NMR (126 MHz, CDCl_3_) δ: 177.1 (C-4), 154.9 (C-8a), 154,5 (t, *^2^J*_C,F_ = 31.3 Hz, C-2) 135.3 (C-7), 126.6 (C-5), 126.3 (C-6), 122.6 (C-4a), 120.8 (t, *^1^J*_C,F_ = 292.5 Hz, CF_2_Cl), 118.4 (C-3), 116.8 (C-8), 42.8 (t, ^4^*J*_C,F_ = 4.5 Hz, CH_2_N_3_); ^19^F-NMR (471 MHz, CDCl_3_) δ: −54.3 (CF_2_Cl); GC-MS, 70 eV, *m*/*z*, (rel. int.): 257 (7) [M − N_2_]^+^, 243 (4) [M − N]^+^, 230 (28) [M − HCN]^+^, 221 (100) [M − Cl]^+^.

*6-Azido-3-methyl-2-trifluoromethylchromone* (**2**). White crystalline solid; yield 89%; mp: 72.5–75.5 °C; UV (MeOH): 203, 230, 318 nm; IR (KBr): 2133, 1649, 1605, 1128 cm^−1^; Raman: 2947, 1650, 1606, 1125 cm^−1^; ^1^H-NMR (500MHz, CDCl_3_) δ: 8.19 (d, *J* = 2.5 Hz, 1H, H-5), 7.69 (d, *J* = 9, 2.5 Hz, 1H, H-7), 7.49 (d, *J* = 8.5 Hz, 1H, H-8), 2.25 (q, ^5^*J*_H,F_ = 2,5 Hz, 3H, CH_3_); ^13^C-NMR (126 MHz, CDCl_3_) δ: 176.81 (C-4), 153.3 (C-8a), 148.5 (s, ^2^*J*_C,F_ = 37.2 Hz, C-2), 136.0 (C-6), 134.9 (C-3), 131,9 (C-4a), 125.4 (C-7), 123.2 (C-8), 120.9 (t, C-2′, ^1^*J*_C,F_ = 280.0 Hz, CF_3_); 119.9 (C-5), 8.7 (q, CH_3_, ^4^*J*_C,F_ = 4.7 Hz); ^19^F-NMR (471 MHz, CDCl_3_) δ: -65.3 (CF_3_); LC-ESI-MS: [M − H]^+^ = 270.834 (calculated: 270.048)

*3-Azidomethyl-2-trifluoromethylchromone* (**3**). White crystalline solid; yield 56%; mp: 80–81 °C; UV (MeOH): 204, 223, 250, 305 nm; IR (KBr): 2104, 1653, 1611, 1166, 897 cm^−1^; Raman: 2107, 1647, 1145, 897 cm^−1^; ^1^H-NMR (300 MHz, CDCl_3_) δ: 8.25 (dd, *J* = 8 and 1.5 Hz, 1H, H-5), 7.79 (ddd, *J* = 8.5, 7 and 1.5 Hz, 1H, H-7), 7.55 (d, *J* = 8.5 Hz, 1H, H-8), 7.50 (ddd, *J* = 8, 7 and 1 Hz, 1H, H-6), 4.46 (q, ^5^*J*_H,F_ = 1 Hz, 2H, CH_2_); ^13^C-NMR (75 MHz, CDCl_3_) δ: 176.9 (C-4), 155.2 (C-8a), 151.0 (q, ^2^*J*_C,F_ = 37.7 Hz, C-2), 135.5 (C-7), 126.8 (C-6), 126.4 (C-5), 122.9 (C-4a), 119.4 (q, ^1^*J*_C,F_ = 277 Hz, CF_3_), 119.0 (q, ^3^*J*_C,F_ = 1 Hz, C-3), 118.5 (C-8), 42.8 (q, ^4^*J*_C,F_ = 2 Hz); ^19^F-NMR (282 MHz, CDCl_3_) δ: -64.5 (CF_3_); GC-MS, 70 eV, *m*/*z*, (rel. int.): 241 (10) [M − N_2_]^+^, 240 (13) [M − H]^+^, 227 (6) [M − N]^+^, 214 (100) [M − HCN]^+^.

*3-Azidomethyl-2-difluoromethylchromone* (**4**). White crystalline solid; yield 70%; mp: 97.5–99.4 °C; UV(MeOH): 204, 224, 246, 301 nm; IR (KBr): 2094, 1657, 1632, 1107, 900 cm^−1^; Raman: 2105, 1659, 1642, 1112, 903 cm^−1^; ^1^H-NMR (600 MHz, CD_3_OD) δ: 8.18 (d, *J* = 8.1 Hz, 1H, H-5); 7.87 (t, *J* = 7.9 Hz, 1H, H-7); 7.66 (d, *J* = 8.5 Hz, 1H, H-8); 7.55 (t, *J* = 7.5 Hz, 1H, H-6); 7.09 (t, ^2^*J*_H,F_ = 51.7 Hz, 1H, CF_2_H); 4.50 (s, 2H, CH_2_N_3_); ^13^C-NMR (151 MHz, CD_3_OD) δ: 178.8 (C-4); 157.2 (t, ^2^*J*_C,F_ = 24.5 Hz, C-2); 157.0 (C-8a); 136.6 (C-7); 127.5 (C-5); 126,7 (C-6); 124.0 (C-4a); 119.8 (t, *^3^J_C,F_* = 3.2 Hz, C-3); 119.6 (C-8); 111.0 (t, ^1^*J*_C,F_ = 241.4 Hz, CF_2_H); 43.3 (CH_2_N_3_); ^19^F-NMR (565 MHz, CD_3_OD) δ: −121.99 (CF_2_H); GC-MS, 70 eV, *m*/*z*, (rel. int.): 223 (38) [M − N_2_]^+^, 209 (14) [M − N]^+^, 203 (100) [M − HF]^+^, 196 (93) [M − HCN]^+^.

*3-Azidomethyl-2-pentafluoroethylchromone* (**5**). Yellow oil at room temperature; yield 28%; UV (MeOH): 204, 221, 246, 304 nm; IR (KBr) 2098, 1659, 1632, 1212, 549 cm^−1^; Raman: 2095, 1657, 1637, 1193, 547 cm^−1^; ^1^H-NMR (600 MHz, CDCl_3_) δ: 8.26 (dd, *J* = 8 and 1.5 Hz, 1H, H-5); 7.80 (ddd, *J* = 8.5, 7.0 and 1,5 Hz, 1H, H-7); 7.56–7.50 (m, 2H, H-6 and H-8); 4.46 (t, ^5^*J*_H,F_ = 2.1 Hz, 2H, CH_2_N_3_); ^13^C-NMR (151 MHz, CDCl_3_) δ: 176,8 (C-4); 155,4 (C-8a); 150,5 (t, ^2^*J*_C,F_ = 27,80 Hz, C-2); 135,4 (C-7); 126,9 (C-5); 126,5 (C-6); 122,8 (C-4a); 121,5 (C-3); 118,4 (C-8); 29,9 (CH_2_N_3_); ^19^F-NMR (565 MHz, CDCl_3_) δ: −83.8 (t, *J* = 2 Hz, 3F, CF_2_CF_3_); −116.1 (q, *J* = 2 Hz, 2F, CF_2_CF_3_); GC-MS, 70 eV, *m*/*z*, (rel. int.): 291 (10) [M − N_2_]^+^, 277 (7) [M − N]^+^, 264 (100) [M − HCN]^+^.

### 3.3. X ray Diffraction Date and Structural Refinement of **4**

The measurements were performed on an Oxford Xcalibur, Gemini, Eos CCD diffractometer (Agilent Technologies XRD Products, Yarnton, UK) with graphite-monochromated MoKα (λ = 0.71073 Å) radiation. X-ray diffraction intensities were collected (ω scans with ϑ and κ-offsets), integrated and scaled with CrysAlisPro [41] suite of programs. The unit-cell parameters were obtained by least-squares refinement (based on the angular setting for all collected reflections with intensities larger than seven times the standard deviation of measurement errors) using CrysAlisPro. Data were corrected empirically for absorption employing the multi-scan method implemented in CrysAlisPro. The non-H structures were solved by the intrinsic phasing procedure implemented in SHELXT [42] and the molecular model refined by full-matrix least-squares with SHELXL of the SHELX suite of programs [43]. The hydrogen atoms were determined in a Fourier difference map phased on the heavier atoms and refined at their found positions with isotropic displacement parameters. Crystal data and structure refinement results are summarized in Appendix A). Crystallographic structural data have been deposited at the Cambridge Crystallographic Data Centre (CCDC, Cambridge, UK). Any request to the Cambridge Crystallographic Data Centre for this material should quote the full literature citation and the reference number CCDC 2119625.

### 3.4. Computational Details

Quantum chemical calculations were performed for the ground state (gas phase) of **1**–**5** with the Gaussian 09 [44]. Scans of the potential energy surface were carried out with the B3LYP/6-311++G(d,p) level of theory. Potential energy curves were performed around the dihedral angles involving the nitrogen, chlorine and fluorine atoms (C3C2-C2′F, C4C3-C3′N, C2C3-C3′N and C2C3-C3′Cl) (see Appendix A).

The geometry optimizations calculations were carried out with the Density Functional Theory (B3LYP) method, employing the 6-311++G(d,p) basis set. In all cases, the calculated vibrational properties correspond to potential energy minima with no imaginary values for the frequencies. The ^1^H- and ^13^C- chemical shifts were calculated for the optimized geometries (B3LYP/6-311+G(2d,p)) using the GIAO method (Gauge Including Atomic Orbital), with the corresponding TMS shielding calculated at the same level of theory. The electronic transitions were calculated with the Time-Dependent Density Functional Theory (TD-DFT), implicitly considering the solvent effect (methanol).

### 3.5. Hirshfeld Surface Calculations of **4**

The 2D-fingerprint plots of the Hirshfeld surface, energy frameworks and lattice interaction energies of **4** were generated using CrystalExplorer v17.5 software [45]. The incidence of each intermolecular interaction in the crystal was visualized and decoded from the 2D-fingerprint plot. The normalized contact distance surface (d_norm_) based on d_i_ and d_e_, (contact distances normalized by the van der Waals (vdW) radii) can be visualized by red spots, the 3D d_norm_ surfaces are mapped over a fixed color scale of −0.243 au (red)–0.824 Å au (blue), shape index in the color range of −1.0 au (concave)–1.0 au (convex) Å and curvedness index in the range of 14−4.0 au (flat)–0.4 au (singular) Å. The surface properties mentioned above (d_norm_, shape and curvedness index) were used to identify planar stacking. Moreover, the intermolecular interaction energies of (**4**) were calculated using TONTO program, integrated in the CrystalExplorer v17.5 software. The interaction energies (Appendix A) between the molecules are obtained using CE-B3LYP model (B3LYP/6-31G(d,p)).

## 4. Conclusions

Five azidochromones were fully characterized by NMR (^1^H-, ^13^C-, ^19^F-) and electronic (UV-Vis) spectroscopies and GC-MS and UHPLC-MS/MS spectrometric methods. From the most stable conformations (theoretical calculations) for each of the compounds, the NMR and electronic spectra were simulated to aid the interpretation and compare these data with experimental results. A good accordance was found between experimental and calculated spectra. The crystal packing of 4 showed weak intermolecular interactions that promote the stabilization of the supramolecular assembly. Therefore, theoretical approaches such as Hirshfeld surface, AIM and NCI analysis were useful to understand the relative contributions of contacts and the character of weak intermolecular interactions in the context of electronic density. The analysis revealed that the F⋅⋅⋅H (23%), N⋅⋅⋅H (22%) and O⋅⋅⋅H (10%) contacts associated to –CF_2_H, –N_3_ and –C=O groups and H⋅⋅⋅H (14%), C⋅⋅⋅C (9%) contacts associated with π···π stacking and C–O···π interactions are the main driving forces in crystal-packing formation. Moreover, the topological parameters studied by NCI/AIM methods evidenced a significant role of the azide (–N=N=N) group contributing through three nitrogen atoms to several bond-critical points. In addition, the reduced density gradient (RDG) characterized the weak intermolecular interactions as van der Waals interactions, where the electronic density can be considered as weak closed-shell (CS) interatomic interactions.

## Data Availability

All data generated or analysed during this study are included in this published article [and its additional information file].

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
