# Peer review of "Synthesis, Experimental and Theoretical Study of Azidochromones"

_molecules, 2022, doi:10.3390/molecules27092636_

Round 1

Reviewer 1 Report

The core structure of chromones represents the important scaffold for synthesis of novel compounds with promising biological activity. The chromones drugs are used in the treatment of allergic diseases, including asthma, allergic rhinitis and systemic mastocytosis.

Although the authors used the known synthetic approach for a relatively small number of new target compounds, the spectral and structural properties of the products have been comprehensively studied by various methods including DFT calculations. The presence of electron-withdrawing polyfluoroalkyl groups attached to the C-2 carbon of chromones can radically modify the biological properties. Taking into account the biological importance of this class of compounds, I propose to accept the paper after correction of following minor flaws.

First of all, in the article it is desirable to give the scheme for the synthesis of new target compounds. In the experimental part, I would advise to write the names of new obtained compounds (as subheadings) starting with a capital letter in italic font. The first ref is better to give in English translation as Russ. Chem. Rev. 2003, 72, 489-516 (the same DOI).

Author Response

Point 1: First of all, in the article it is desirable to give the scheme for the synthesis of new target compounds.

Point 2: In the experimental part, I would advise to write the names of new obtained compounds (as subheadings) starting with a capital letter in italic font.

Point 3: The first ref is better to give in English translation as Russ. Chem. Rev. 2003, 72, 489-516 (the same DOI).

Response 1: Taking into account the reviewer's suggestion, the following scheme was added, in section 3.2.

Scheme 2 (Review the attached document)

Reviewer 2 Report

The paper describes the synthesis, experimental and theoretical study of some azidochromones. The products described are novel and their characterisation is complete. The work is moderately interesting.
I suggest that the paper is accepted if the following issues are corrected.

Corrections/comments:
-Please include a reaction scheme in the manuscript showing the synthesis of the target compounds.
Experimental:
-Please provide the logE values for the UV absorptions.

Author Response

Response 1: Taking into account the reviewer's suggestion, the following scheme was added, in section 3.2.

Scheme 2 (Review the attached document)

Response 2: In Table 3, the logE value for UV absorptions was added in parentheses. Likewise, at the end of the table, a description of the value is given in parentheses.

               Table 3. Experimental and calculated (B3LYP/6-311++G(d,p) electronic spectra of 1 – 5.

Compound

Experimentala

Calculatedb

Assignment

1

204 (4.62)

211 (0.177)

HOMO-1 → LUMO+3 (51%)

225 (4.42)

237 (0.136)

HOMO-5 → LUMO (49%)

244 (4.23)

289 (0.121)

HOMO-2 → LUMO (88%)

303 (4.00)

330 (0.076)

HOMO → LUMO (74%)

2

203 (3.87)

212 (0.228)

HOMO → LUMO+5 (25%)

230 (3.81)

271 (0.599)

HOMO-2 → LUMO (41%)

318 (3.24)

361 (0.116)

HOMO → LUMO (88%)

3

204 (4.55)

201 (0.185)

HOMO-2→LUMO+2 (43%)

223c (4.34)

223 (0.103)

HOMO-5→LUMO+1 (50%)

250c (4.06)

250 (0.230)

HOMO→LUMO+1(37%)

305 (3.95)

304 (0.091)

HOMO→LUMO (95%)

4

204 (4.63)

204 (0.306)

HOMO-2→LUMO+2 (44%)

224 (4.56)

233 (0.140)

HOMO→LUMO+2 (37%)

HOMO-4→LUMO (32%)

246c (4.36)

250 (0.210)

HOMO→LUMO+1 (64%)

301 (4.14)

300 (0.093)

HOMO→LUMO (80%)

5

204 (4.65)

203 (0.202)

HOMO-2→LUMO+2 (50%)

221 (4.46)

223 (0.079)

HOMO-5→LUMO (59%)

246 (4.35)

238 (0.108)

HOMO-4→LUMO (72%)

249 (0.233)

HOMO→LUMO+1 (65%)

271 (0.084)

HOMO-2→LUMO (75%)

304 (4.12)

305 (0.108)

HOMO→LUMO (82%)

a In nm and logε (in parentheses) . b In nm and oscillator strength (in parentheses) in a.u. c Shoulder.

Reviewer 3 Report

The manuscript reported a series of 2-(haloalkyl)-3-azidomethyl and 6-azido chromone compounds that were synthetized, characterized and studied by theoretical and spectroscopic methods. The crystal structure of compound 4 was determined by X-ray diffraction method. The quality of structure data is pretty good. Although the similar compounds were studied and revealed by the authors and other people in this field in literatures, the current research reported in this manuscript is quite interesting and a good addition to this field with quite thorough characterizations. I think that the manuscript is suitable for publication in Molecules with minor revisions. There are some typos that the authors should consider to make corrections, for example,

In the manuscript page1, in line 35, “… crystal packing thought …”, the “thought” should be “through”.

Author Response

Point 1: In the manuscript page1, in line 35, “… crystal packing thought …”, the “thought” should be “through”.

Response 1: Done
